# Aorto-Iliac Artery Calcification Prior to Kidney Transplantation

**DOI:** 10.3390/jcm9092893

**Published:** 2020-09-07

**Authors:** Stan Benjamens, Elsaline Rijkse, Charlotte A. te Velde-Keyzer, Stefan P. Berger, Cyril Moers, Martin H. de Borst, Derya Yakar, Riemer H. J. A. Slart, Frank J. M. F. Dor, Robert C. Minnee, Robert A. Pol

**Affiliations:** 1Department of Surgery, Division of Transplant Surgery, University of Groningen, University Medical Center Groningen, 9713 GZ Groningen, The Netherlands; c.moers@umcg.nl (C.M.); r.pol@umcg.nl (R.A.P.); 2Medical Imaging Center, Department of Nuclear Medicine and Molecular Imaging & Department of Radiology, University of Groningen, University Medical Center Groningen, 9713 GZ Groningen, The Netherlands; d.yakar@umcg.nl (D.Y.); r.h.j.a.slart@umcg.nl (R.H.J.A.S.); 3Department of Surgery, Division of HPB and Transplant Surgery, Erasmus MC University Medical Center, 3015 CE Rotterdam, The Netherlands; a.rijkse@erasmusmc.nl (E.R.); r.minnee@erasmusmc.nl (R.C.M.); 4Department of Internal Medicine, Division of Nephrology, University of Groningen, University Medical Center Groningen, 9713 GZ Groningen, The Netherlands; c.a.keyzer@umcg.nl (C.A.t.V.-K.); s.p.berger@umcg.nl (S.P.B.); m.h.de.borst@umcg.nl (M.H.d.B.); 5Department of Biomedical Photonic Imaging, Faculty of Science and Technology, University of Twente, 7522 NB Enschede, The Netherlands; 6Imperial College Renal and Transplant Centre, Hammersmith Hospital, Imperial College Healthcare NHS Trust, London W12 0HS, UK; frank.dor@nhs.net; 7Department of Surgery & Cancer, Imperial College, London SW7 2BU, UK

**Keywords:** kidney transplantation, vascular calcification, aorta, iliac artery, cardiovascular diseases

## Abstract

As vascular calcification is common in kidney transplant candidates, aorto-iliac vessel imaging is performed for surgical planning. The aim of the present study was to investigate whether a novel non-contrast enhanced computed tomography-based quantification technique for aorto-iliac calcification can be used for cardiovascular risk stratification prior to kidney transplantation. In this dual-center cohort study, we measured the aorto-iliac calcium score (CaScore) of 547 patients within three years prior to transplantation (2005–2018). During a median (interquartile range) follow-up of 3.1 (1.4, 5.2) years after transplantation, 80 (14.7%) patients died, of which 32 (40.0%) died due to cardiovascular causes, and 84 (15.5%) patients had a cardiovascular event. Kaplan-Meier survival curves showed significant differences between the CaScore tertiles for cumulative overall-survival (Log-rank test *p* < 0.0001), cardiovascular survival (*p* < 0.0001), and cardiovascular event-free survival (*p* < 0.001). In multivariable Cox regression, the aorto-iliac CaScore was associated with all-cause mortality (hazard ratio 1.53, 95%CI 1.14–2.06, *p* = 0.005), cardiovascular mortality (2.04, 1.20–3.45, *p* = 0.008), and cardiovascular events (1.35, 1.01–1.80, *p* = 0.042). These independent associations of the aorto-iliac CaScore with the outcome measures can improve the identification of patients at risk for (cardiovascular) death and those who could potentially benefit from stringent cardiovascular monitoring to improve their prognosis after transplantation.

## 1. Introduction

Kidney transplantation improves survival of patients with end-stage renal disease (ESRD) and reduces cardiovascular disease burden [1,2]. Despite the improved prognosis after transplantation, patients remain vulnerable, with cardiovascular disease as the leading cause of death [2,3]. A high overall prevalence of cardiovascular disease in the first year after kidney transplantation is reported, with myocardial infarction (MI) in 5.1%, cerebrovascular accident (CVA)/transient ischemic attack (TIA) in 7.3%, and peripheral artery disease (PAD) in 15.8% of patients [2,4].

Multidisciplinary evaluation, including consultation of a vascular surgeon, to evaluate the cardiovascular status of kidney transplant candidates could improve the identification of patients at risk for (cardiovascular) death and those who could potentially benefit from interventions to improve their prognosis after transplantation [5,6,7]. When screening kidney transplant candidates, evaluation generally includes traditional risk factors, as used in risk prediction scores for the general population (i.e., the Framingham score), and transplant candidate-specific risk factors, with among others chronic kidney disease–mineral bone disorder (CKD-MBD) [8,9]. As ESRD accelerates vascular calcification and kidney transplantation incompletely decreases this acceleration, cardiovascular risk stratification is not always straightforward in this population [10,11].

Prior studies have identified imaging-proven vascular calcification as a prognostic factor for inferior survival and cardiovascular events in kidney transplant recipients [12,13]. For the applied imaging studies, a distinction can be made between coronary artery calcification (CAC) and aorto-iliac calcification, as well as between assessment of calcification prior to and after transplantation. CAC, expressed as Agatston score by non-contrast enhanced cardiac computed tomography (CT), demonstrated progression within four years in prevalent kidney transplant recipients and is associated with myocardial infarction and mortality [14,15,16,17,18]. Aorto-iliac calcification, by plain pelvic X-ray, dual X-ray absorptiometry, or semi-quantitative CT assessment, was associated with inferior overall patient survival [12,19,20,21,22,23,24].

To date, CT imaging of the aorto-iliac vessels prior to kidney transplantation is primarily performed for surgical planning, while the available imaging data are not widely used for cardiovascular risk stratification in clinical practice [25,26]. As the imaging data are available, non-contrast enhanced CT-based quantification of aortic calcification can be used for cardiovascular risk stratification of patients screened for kidney transplantation, without the need for additional procedures [27,28].

Here, we present the results of a dual-center cohort study for the quantification of aorto-iliac calcification prior to kidney transplantation and its association with all-cause mortality, cardiovascular mortality, and cardiovascular events, using the aorto-iliac calcium score (CaScore)—an adapted Agatston score.

## 2. Methods

### 2.1. Subjects and Study Design

Patients referred for a non-contrast enhanced CT-procedure, as part of the screening of high-risk patients prior to kidney transplantation, were included in a dual-center cohort study (University Medical Center Groningen and Erasmus MC University Medical Center, The Netherlands). In line with the pre-transplant screening protocol in both transplant centers, a CT-procedure was required for transplant screening if either of the following were present: age > 50 years, dialysis vintage > 2 years, a history of PAD or signs and/or symptoms of PAD, diabetes mellitus, or prior surgery in the iliac fossa. Patients were referred for pre-transplant cardiac evaluation if the following were present: symptoms and signs of cardiac disease, a history of cardiac disease, reduced exercise tolerance, PAD, or diabetes mellitus. All adult patients undergoing kidney transplantation between January 2005 to December 2018, in which a CT-procedure was performed within three years prior to transplantation, were considered eligible. Transplant-related data was extracted from the Dutch Organ Transplant Registry and the electronic patient records were screened for baseline and follow-up variables. The study was performed in line with the Declaration of Helsinki and the principles outlined in the Declaration of Istanbul on Organ Trafficking and Transplant Tourism. The Medical Ethical Committee gave approval (University Medical Center Groningen 2017/523) for this dual-center cohort study.

### 2.2. Outcome Measures and Clinical Variables

The primary outcome measure of this study was all-cause mortality and the secondary outcome measures were cardiovascular mortality and cardiovascular events (including cardiovascular mortality and events). Cardiovascular events were defined as MI (International Statistical Classification of Diseases and Related Health Problems (ICD)-10: I21), CVA (ICD I60-I66), TIA (ICD-10 G45), or new onset PAD (ICD-10 I73) Fontaine III or IV. Cardiovascular mortality was defined as death due to a cardiovascular event or heart failure (ICD-10 I50). Baseline characteristics consisted of the following covariables at time of transplantation: gender, age, diabetes mellitus (fasting glucose ≥ 7.0 mmol/L, casual plasma glucose ≥ 11.1 mmol/L + diabetes symptoms, or glucose tolerance test with 2-h plasma glucose ≥ 11.1 mmol/L), Body Mass Index (BMI in kg/m^2^), smoking status (non, former, current), total cholesterol level (mmol/L), hypercholesterolemia (total cholesterol level > 5.2 mmol/L or use of lipid lower medication), systolic and diastolic blood pressure (mmHg), and use of antihypertensive medication. History of cardiovascular disease consisted of MI, CVA, TIA, or PAD. Comorbidities were scored using the Charlson Comorbidity Index and the 10-year risk for MI was estimated with the Framingham score [8,29]. Transplant-related data consisted of date of transplantation, type of donation (living-donation, donation after circulatory death, and donation after brain death), number of previous transplantations, type of dialysis (pre-emptive, hemodialysis, or peritoneal dialysis), and dialysis vintage pre-transplantation. Data at one-year post-transplant consisted of estimated-Glomerular Filtration Rate (eGFR), calcium, phosphate, albumin, parathyroid hormone (PTH), and use of calcineurin inhibitors (cyclosporin or tacrolimus).

### 2.3. Quantification of Aorto-iliac Calcification

Non-contrast enhanced CT images of the aorto-iliac trajectory were extracted from the imaging databases of both hospitals. Quantification of aorto-iliac calcification was performed for three vascular segments, being the abdominal aorta inferior of the renal arteries, the common iliac artery, and the external iliac artery on the side of the subsequent transplant, by an examiner blinded for the patients’ medical details (Figure 1).

The selected areas of calcification were reviewed on the axial, sagittal, and coronals views, and adjusted if necessary [28]. Using the syngo.CT CaScoring software (Siemens Healthineers, Erlangen, Germany), with the standard calcification threshold of 130 Hounsfield Units (HU), the Agatston score adapted for the aorto-iliac trajectory (aorto-iliac CaScore) was obtained for the three segments [27].

### 2.4. Statistical Analysis

Based on the aorto-iliac CaScore, patients were stratified into tertiles of equal size: low CaScore (0–859 HU), medium CaScore (860–5600 HU), and high CaScore (>5600 HU). The data are presented as mean (standard deviation, (SD)) in case of normal distribution, median (interquartile range, (IQR)) for skewed data, and number (percentage, %) for categorical data, for the full cohort and the aorto-iliac CaScore tertiles. Tertiles were compared using a chi-square test or one-way analysis of variance. The median (IQR) time of follow-up was calculated with the reversed Kaplan-Meier method, with the date of transplantation as the start of the follow-up. The cumulative overall, event-free survival, and cardiovascular survival were calculated using Kaplan-Meier survival curves, comparing aorto-iliac CaScore tertiles with Log-rank testing. Univariable and multivariable Cox proportional hazards regression analysis were used to evaluate the association of the aorto-iliac CaScore, as a categorical and continuous variable, with all-cause mortality, cardiovascular events, and cardiovascular mortality (Appendix A). Additional survival analysis was performed for time between transplantation and death with a functioning graft (censored for return to dialysis and re-transplantation). To avoid overfitting of the multivariable model with the a priori selected covariables, the results are presented in six steps, as presented in Table 4. Additional analyses of the interaction with covariables was performed by univariable subgroup analysis and analysis of interaction terms for aorto-iliac CaScore (continuous) and all-cause mortality, with groups based on the mean or median for continuous variables. The *p*-values of interaction terms were considered significant when <0.05 (Keyzer et al.) or <0.20 (Ramos et al.) [11,30]. Skewed variables were transformed to the natural log (log(x + 1)) for the regression analyses and results are presented as hazard ratio (HR) and corresponding 95% confidence interval (95% CI). The additional discriminative capacity of the aorto-iliac CaScore compared to a basic model (gender and age), the Charlson Comorbidity Index, and the Framingham score was assessed using C-statistics (Harrell’s concordance index), a change in C-statistics, and the integrated discrimination improvement (IDI). The IDI is a statistical method to describe the predictive performance of biomarkers in extensive regression models, by comparing an initial model with a model including this new biomarker [31]. A *p*-value of <0.05 was considered statistically significant. Statistical analyses were performed with R: A Language and Environment for Statistical Computing, version 1.0.153 for Mac (R Foundation for Statistical Computing, Vienna, Austria).

## 3. Results

### 3.1. Study Population

This dual-center cohort consisted of 547 kidney transplant recipients, with a mean time between CT and transplantation of 0.91 (0.72) years. In this cohort, 336 (61.4%) patients were male, the median age at time of transplantation was 60 (51, 68) years, 173 (31.6%) patients had diabetes mellitus, and 350 (64.0%) were on dialysis. Additional baseline characteristics are summarized in Table 1, variables at one-year post-transplant are summarized in Table 2.

### 3.2. Aorto-Iliac CaScore and All-Cause Mortality

After a median follow-up of 3.1 (1.4, 5.2) years after transplantation, 80 (14.7%) patients had died (Table 3). The Kaplan-Meier survival curve for overall survival showed a significant difference between the tertiles (*p* < 0.0001) (Figure 2).

In univariable Cox regression analysis, the continuous CaScore was significantly associated with all-cause mortality (HR 2.02, 95% CI 1.49–2.56, *p* < 0.0001), with consistent HRs in the subgroup analysis (Table 4 and Figure 3). Based on the p-value cut-off by Keyzer et al. (<0.05), no significant interaction terms were found. With the cut-off by Ramos et al. (<0.20), the variables age (*p* = 0.191) and diabetes mellitus (*p* = 0.141) showed a significant interaction. In multivariable Cox regression analysis, the association between the CaScore (continuous) and all-cause mortality remained significant after adjustment in six models, with HR 1.53 (95% CI 1.14–2.06, *p* = 0.005) in model 5 and HR 1.72 (95% CI 1.22–2.43, *p* = 0.002) in model 6. A considerably higher HR was observed for the high CaScore (HR 2.59, 95% CI 1.22–5.51) compared to the medium CaScore (HR 1.27, 95% CI 0.60–2.66), with the low CaScore as reference, with P(trend) = 0.004 (model 5) (Table 4). When censoring patients with graft failure, the association between the CaScore (continuous) and all-cause mortality (*n* = 47) remained significant, with HR 2.32 (95% CI 1.52–3.55, *p* < 0.0001) after adjustment in five models.

### 3.3. Aorto-iliac CaScore and Cardiovascular Mortality

During the follow-up period, in 32 (40.0%) out of the 80 patients who died, mortality was due to cardiovascular causes (Table 3). For the remaining patients who died: 17 (21.2%) out of 80 died due to infection, 15 (18.8%) due to an oncologic cause, three (3.8%) due to pulmonary disease, and for 13 (16.2%), the cause of death could not be established. The Kaplan-Meier survival curve for cardiovascular survival showed a significant difference between the tertiles (*p* < 0.0001) (Figure 4).

In univariable Cox regression analysis, a significant association with cardiovascular mortality was observed for the continuous CaScore (HR 2.27, 95% CI 1.55–3.34, *p* < 0.0001). In multivariable Cox regression analysis, the association between the CaScore and cardiovascular mortality remained significant after adjustment in six models, HR 2.04 (95% CI 1.20–3.45, *p* = 0.008) (continuous) in model 5 and HR 2.13 (95% CI 1.32–3.43, *p* = 0.002) in model 6. A considerably higher HR was observed for the high CaScore (HR 4.46, 95% CI 1.19–16.71) compared to the medium CaScore (HR 1.64, 95% CI 0.46–5.76), with the low CaScore as reference, with P(trend) = 0.014 (model 5) (Table 4).

### 3.4. Aorto-Iliac CaScore and Cardiovascular Events

During the follow-up period, a total of 84 (15.5%) patients had one or more cardiovascular events, as described in detail in Table 3. The Kaplan-Meier survival curve for cardiovascular event-free survival showed a significant difference between the tertiles (*p* < 0.001) (Figure 4).

In univariable Cox regression analysis, a significant association with cardiovascular events was observed for the continuous CaScore (HR 1.56, 95% CI 1.25–1.95, *p* < 0.0001). In multivariable Cox regression analysis, the association between the CaScore and cardiovascular events remained significant after adjustment in six models, HR 1.35 (95% CI 1.01–1.80, *p* = 0.042) (continuous) in model 5 and HR 1.45 (95% CI 1.09–1.92, *p* = 0.010) in model 6. No significant increase in HR was observed for the high CaScore (HR 1.98, 95% CI 0.97–4.07) compared to the medium CaScore (HR 1.47, 95% CI 0.76–2.85), with the low CaScore as reference, with P(trend) = 0.060 (model 6) (Table 4).

### 3.5. Discriminative Capacity of the Aorto-iliac CaScore

The aorto-iliac CaScore alone had a C-statistic of 0.66 (95% CI 0.49-0.72) for all-cause mortality. When added to a basic model of gender and age, a significant improvement of the c-statistic (0.07, 95% CI 0.01–0.14, *p* = 0.042) and a positive and significant IDI (4.1%, *p* < 0.0001) were observed. Addition of the CaScore to the Charlson Comorbidity Index and the Framingham score, resulted in c-statistics of 0.75 (95% CI 0.67–0.83) and 0.76 (95% CI 0.69–0.83), respectively. The improvement in c-statistics were not significant, whereas the IDI’s were positive and significant for the addition of the CaScore to the Charlson Comorbidity Index (IDI 4.2, *p* < 0.0001) and the Framingham score (IDI 2.5, *p* < 0.001) (Appendix A). C-statistics of the CaScore alone for cardiovascular mortality and cardiovascular events were 0.72 (95% CI 0.63–0.81) and 0.58 (95% CI 0.52–0.64), respectively. Changes in c-statistics and IDI’s for these outcome measures are presented in Appendix A.

## 4. Discussion

Non-contrast enhanced CT-based quantification of the aorto-iliac vessels during screening prior to kidney transplantation, as an adapted Agatston score, is a novel approach in transplant medicine. In the kidney transplant recipient cohort presented, the aorto-iliac CaScore showed to be associated with all-cause and cardiovascular mortality, and cardiovascular events after adjustment for traditional and transplant candidate-specific risk factors. The identified association was consistent in subgroup and interaction analyses, and the CaScore showed to have a significant discriminative capacity for all-cause and cardiovascular mortality.

The baseline differences, as compared for the aorto-iliac CaScore tertiles, are in line with previous studies on cardiovascular risk stratification prior to kidney transplantation [2]. The traditional risk factors, i.e., male gender, age, BMI, current or former smoker, hypercholesterolemia, and hypertension, and the transplant candidate-specific risk factor dialysis vintage pre-transplantation, were more prevalent in the medium and high CaScore tertiles. To note, there were no differences between the tertiles regarding diabetes mellitus prior to transplantation. The relatively high Charlson Comorbidity Index pointed towards more comorbidities and the Framingham score showed more expected cardiovascular events in the medium and high CaScore tertiles. The incidence of cardiovascular events was high (*n* = 84, 15.5%) compared to previously published outcomes of kidney transplant recipients in the Netherlands (8.5–11%, in 5.4 and 7 years, respectively), which was expected since the indication for a pre-transplant CT-scan was defined by cardiovascular risk factors [12,32].

To establish the clinical relevance of the aorto-iliac CaScore, a comparison was made with clinical variables available at the time of screening prior to transplantation. The discriminative capacity of the Charlson Comorbidity Index and Framingham score for all-cause and cardiovascular mortality significantly increased when the CaScore was added. The c-statistics and IDI’s of the CaScore for cardiovascular events were considerably lower compared to those for all-cause and cardiovascular mortality.

The quantification technique for vascular calcification is considered to be reproducible and reliable, with an interobserver agreement of ≥0.99 and an inter-scanner agreement of ≥0.97 [33,34]. In this study we focused on the total calcium volume in the aorto-iliac trajectory; the advantages of this approach are the quantification after non-contrast enhanced CT procedures and the use of CT procedures with varying slice-thicknesses [27]. In the general population, application of this technique resulted in a significant difference in five-year cardiovascular event-free survival between the quartiles of abdominal aorta calcification, outperforming the Framingham score (area under the curve of 0.76 versus 0.64, *p* = 0.013) [28].

Two relatively small studies focused on non-contrast enhanced aorto-iliac CT prior to kidney transplantation, the first (*n* = 131) with a 14-point visual grading scale and the second with quartiles of calcification, demonstrating correlations with surgical complexity and cardiovascular surgery during follow-up [23,24]. In contrast to the independent association identified in the current study, the two earlier studies did not find significant associations with all-cause mortality analyses, in cohorts with 21 and four events, respectively. The results of a more recent study, focusing on aorto-iliac stenosis on contrast-enhanced CT prior to transplantation and all-cause mortality, were in line with our findings for all-cause mortality [22].

The literature on abdominal aorta calcification (AAC) assessment by plain pelvic X-ray or dual X-ray absorptiometry, using visual grading scales, is more extensive [13]. Studies evaluating this method showed results similar to our findings, with significant associations of a high-AAC with all-cause mortality and cardiovascular events in multivariable Cox regression analyses [12,20,21]. Timing of the imaging procedure in previous studies is an important difference compared with the current study, as imaging procedures were performed after transplantation and at the physician’s discretion, instead of CT procedures during the screening prior to transplantation.

For CAC prior to and after kidney transplantation, identified by non-contrast CT, several studies have shown the association with cardiovascular events and mortality [17,18]. The first prospective study showing the association of CAC after transplantation and cardiovascular events is from Nguyen et al., with a mean follow-up of 2.3 years and a total of 31 events [17]. In a small study (*n* = 112) with asymptomatic transplant recipients, Roe et al. showed that laboratory markers of inflammation are associated with the severity of CAC [18].

To our knowledge, this is the largest kidney transplant recipient cohort with CT imaging prior to transplantation available for re-evaluation. Strengths of this study include the median follow-up of 3.1 years, the representative case-mix of pre-emptive (36.0%) and dialysis-dependent (64.0%) patients, and the relatively large number of outcome measures studied for all-cause mortality, cardiovascular mortality, and cardiovascular events, compared to earlier studies. Additionally, the quantification technique for vascular calcification is considered to be reproducible and reliable, with an interobserver agreement of ≥0.99 and an inter-scanner agreement of ≥0.97 [33]. For the number of included variables in the multivariable analyses, we followed the recommendation by Vittinghoff and McCulloch of 5–9 events per predictor variable [35]. This recommendation could be followed for the outcome measures all-cause mortality and cardiovascular events; however, there is a risk for overfitting in model 4 and model 5 with the outcome measure cardiovascular mortality. The main limitation of this study is the selection of patients based on the availability of CT images prior to transplantation, resulting in a cohort of patients with a relatively high age at transplantation (median 60 years), more comorbidities (mean Charlson Comorbidity Index of 6.4), and a subsequent higher mortality risk. While the Charlson Comorbidity Index was high compared to a previously published Dutch kidney transplant cohort, with a mean of 2.8 (1.0), the median Framingham score of 9.6 (3.4, 18.8) was comparable to earlier studies [8,36]. The in- and exclusion criteria limit conclusions regarding the assessment of patient’s suitability for transplantation, as the results of this study cannot be generalized to the overall transplant candidate population. As the tertiles are based on equal-size groups, the reported CaScore ranges per tertile are highly population dependent. External validation of these CaScore tertile ranges is therefore required.

## 5. Conclusions

In conclusion, in this dual-center kidney transplant recipient cohort, we demonstrated an independent association of the aorto-iliac CaScore, extracted from commonly performed non-contrast enhanced CT procedures prior to transplantation, with all-cause mortality, cardiovascular mortality, and cardiovascular events. These findings can improve the identification of patients at risk for (cardiovascular) death and those who could potentially benefit from stringent cardiovascular monitoring to improve their prognosis after transplantation.

## Figures and Tables

**Figure 1 jcm-09-02893-f001:**
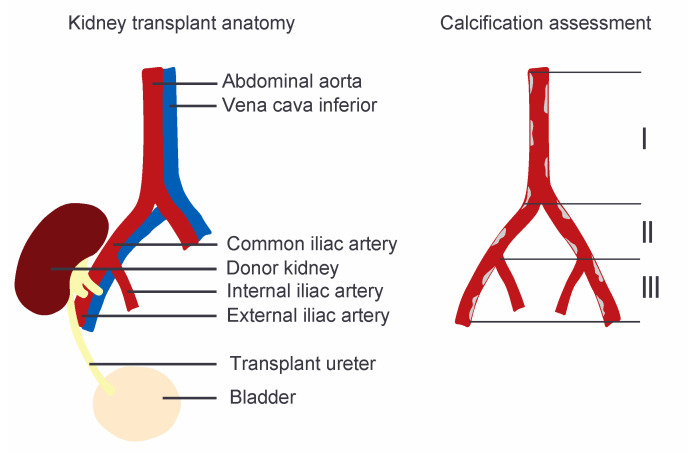
Graphical image of the kidney transplant anatomy and the aorto-iliac calcium score (CaScore) assessment performed in three vascular segments, being (I) the abdominal aorta inferior of the renal arteries, (II) the common iliac artery, and (III) the external iliac artery on the side of the subsequent transplant.

**Figure 2 jcm-09-02893-f002:**
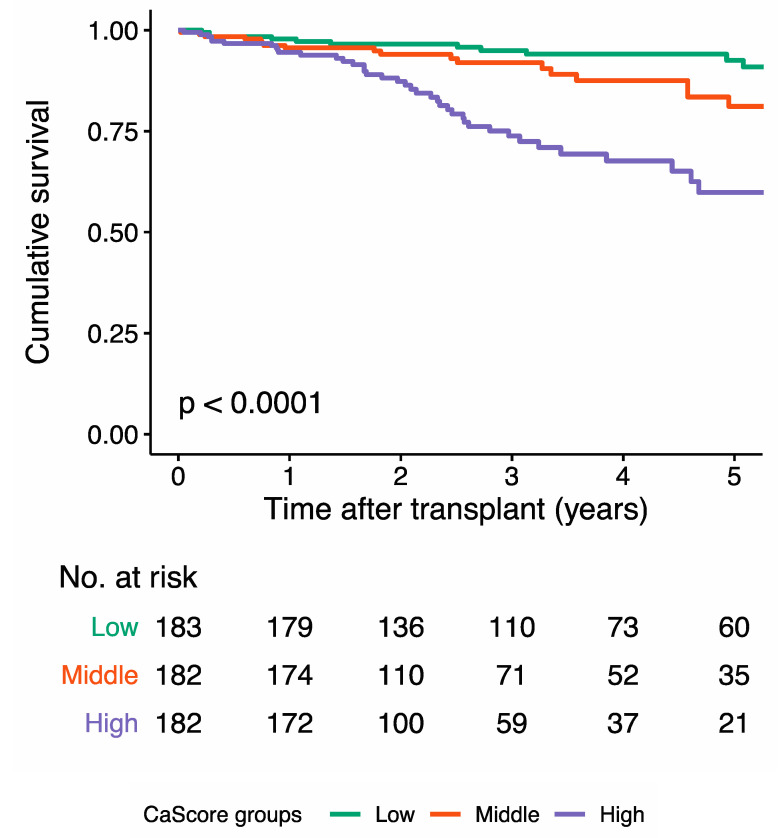
Kaplan-Meier curve for cumulative overall survival for the low aorto-iliac calcium score (CaScore tertile), the medium CaScore tertile, and the high CaScore tertile. The overall survival at the median follow-up time was 94.9% (95% CI 91.5–98.5) in the low, 90.5% (95% CI 85.4–95.9) in the medium, and 71.0% (95% CI 62.7–80.3) in the high CaScore tertile.

**Figure 3 jcm-09-02893-f003:**
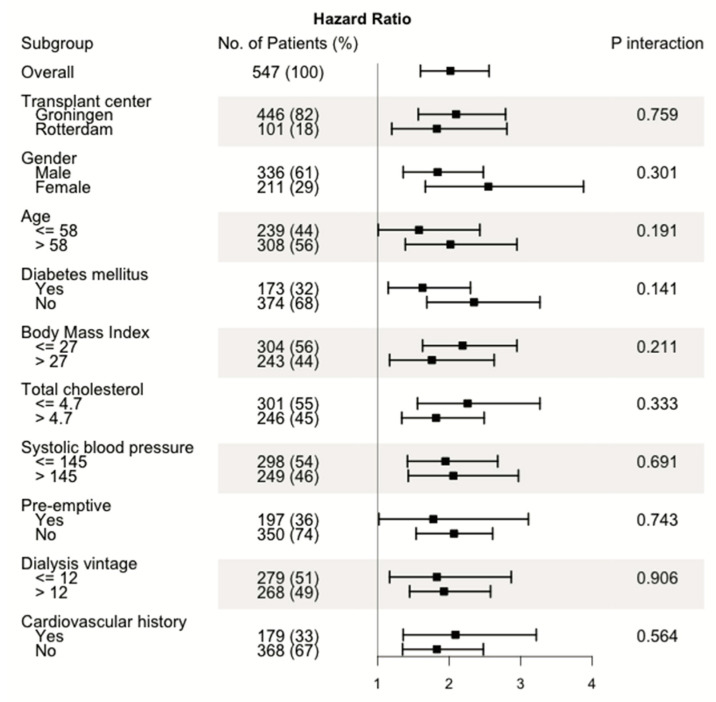
Forest plot of hazard ratios for the univariable subgroup analysis and *p*-values of interaction analysis for aorto-iliac calcium score (CaScore) (continuous) and all-cause mortality.

**Figure 4 jcm-09-02893-f004:**
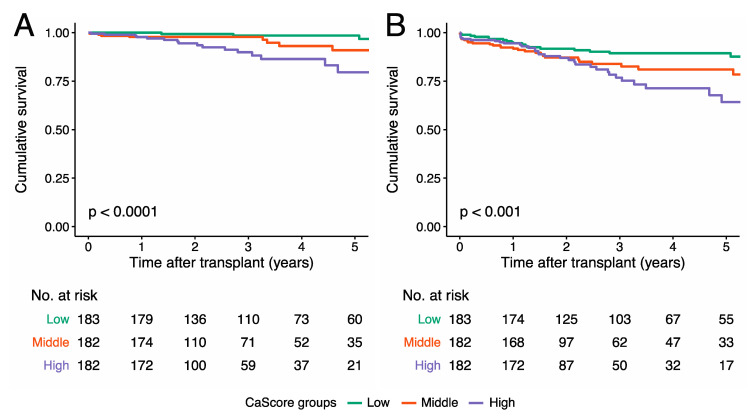
Kaplan-Meier curve for (**A**) cardiovascular survival and (**B**) cardiovascular event-free survival for the low aorto-iliac CaScore tertile, the medium CaScore tertile, and the high CaScore tertile. The cardiovascular survival at the median follow-up time was 98.5% (95% CI 96.4–100.0) in the low, 96.3% (95% CI 92.8–99.9) in the medium, and 86.4% (95% CI 79.5–94.0) in the high CaScore tertile. The cardiovascular event-free survival at the median follow-up time was 89.3% (95% CI 84.6–94.4) in the low, 81.0% (95% CI 74.2–88.5) in the medium, and 75.2% (95% CI 67.0–85.5) in the high CaScore tertile.

**Table 1 jcm-09-02893-t001:** Patient characteristics.

Variables	Total (*n* = 547)	Low (*n* = 183)	Medium (*n* = 182)	High (*n* = 182)	*p*-Value
Male gender ^a^	336 (61.4)	98 (53.6)	108 (57.8)	131 (72.0)	**0.001** ^d^
Age, years ^c^	60 (51, 68)	48 (39, 56)	62 (55, 68)	66 (60, 71)	**<0.001** ^e^
Diabetes mellitus ^a^	173 (31.6)	56 (30.6)	61 (33.5)	56 (30.8)	0.798 ^d^
Body Mass Index, kg/m^2 b^	26.8 (4.7)	26.0 (5.1)	27.7 (4.6)	26.7 (4.1)	**0.002** ^e^
Smoker ^a^					**<0.001** ^d^
Non	154 (28.2)	74 (40.4)	45 (24.7)	35 (19.2)	
Former	332 (60.7)	89 (48.6)	117 (64.3)	126 (69.2)	
Current	61 (11.2)	20 (10.9)	20 (11.0)	21 (11.5)	
Total cholesterol, mmol/L ^b^	4.7 (1.3)	4.7 (1.2)	4.8 (1.5)	4.5 (1.2)	0.066 ^e^
Hypercholesterolemia ^a^	153 (28.0)	37 (20.2)	59 (32.4)	57 (31.3)	**0.016** ^d^
Systolic blood pressure, mmHg ^b^	145 (22)	141 (20)	145 (22)	148 (24)	**0.014** ^e^
Diastolic blood pressure, mmHg ^b^	80 (14)	82 (13)	80 (13)	78 (14)	**0.018** ^e^
Use of antihypertensive medication ^a^	439 (80.3)	132 (72.1)	150 (82.4)	157 (86.3)	**0.002** ^d^
Type of donation ^a^					0.200 ^d^
Living-donation	301 (55.0)	110 (60.1)	105 (57.7)	86 (47.3)	
Donation after circulatory death	128 (23.4)	39 (21.3)	36 (19.8)	53 (29.1)	
Donation after brain death	118 (21.6)	34 (18.6)	41 (22.5)	43 (23.6)	
Previous transplants ^a^					0.161 ^d^
Non	514 (94.0)	167 (93.3)	176 (96.7)	171 (94.0)	
One	22 (4.0)	10 (5.5)	4 (2.2)	8 (4.4)	
Two	11 (2.0)	6 (3.3)	2 (1.1)	3 (1.6)	
Type of dialysis ^a^					0.196 ^d^
Pre-emptive	197 (36.0)	71 (38.8)	71 (39.0)	55 (30.2)	
Hemodialysis	250 (45.7)	77 (42.1)	85 (46.7)	88 (48.4)	
Peritoneal dialysis	100 (18.3)	36 (19.1)	26 (14.3)	39 (21.4)	
Dialysis vintage pre-transplantation, months ^c^	12 (0, 28)	10 (0, 23)	12 (0, 29)	16 (0, 33)	**0.017** ^f^
History of MI ^a^	82 (15.0)	9 (4.9)	30 (16.5)	43 (23.6)	**<0.001** ^d^
History of CVA ^a^	40 (7.3)	7 (3.8)	11 (6.0)	22 (12.1)	**0.007** ^d^
History of TIA ^a^	23 (4.2)	4 (2.2)	11 (6.0)	8 (4.4)	0.183 ^d^
History of PAD ^a^	78 (14.3)	18 (9.8)	24 (13.2)	36 (19.8)	**0.022** ^d^
Charlson Comorbidity Index ^b^	6.4 (2.5)	5.1 (2.0)	6.3 (1.9)	7.7 (2.7)	**<0.001** ^e^
Framingham score, % ^c^	9.6 (3.4, 18.8)	3.5 (0.6, 9.5)	10.2 (4.1, 20.2)	16.6 (9.4, 23.8)	**<0.001** ^f^

MI = myocardial infarction; CVA = cerebrovascular accident; TIA = transient ischemic attack; PAD = peripheral artery disease; ^a^ number (%); ^b^ mean ± standard deviation (SD); ^c^ median (interquartile range); ^d^
*p*-value by chi-square test; ^e^
*p*-value by One-way ANOVA; ^f^
*p*-value by Kruskal-Wallis test; *p*-values in bold indicate statistical significance.

**Table 2 jcm-09-02893-t002:** One-year post-transplant variables.

Variables	Total (*n* = 547)	Low (*n* = 183)	Medium (*n* = 182)	High (*n* = 182)	*p*-Value
Laboratory values ^a^					
eGFR ^b^	51.3 (20.6)	55.9 (22.2)	50.2 (18.1)	47.68 (20.82)	**0.001** ^e^
Calcium, mmol/L ^b^	2.43 (0.14)	2.40 (0.14)	2.45 (0.13)	2.45 (0.14)	**0.001** ^e^
Phosphate, mmol/L ^b^	0.93 (0.21)	0.94 (0.23)	0.95 (0.19)	0.89 (0.19)	**0.014** ^e^
Albumin, g/L ^b^	43.3 (3.1)	43.2 (3.4)	43.6 (2.9)	43.0 (3.0)	0.237 ^e^
PTH, pmol/L ^c^	10.4 (7.4, 15.6)	9.4 (6.5, 12.8)	10.6 (7.8, 16.3)	11.6 (8.0, 16.4)	**0.020** ^f^
CNI ^d^	535 (97.8)	178 (97.3)	179 (98.4)	178 (97.8)	0.779 ^g^
Ciclosporin ^d^	42 (7.7)	16 (8.7)	13 (7.1)	13 (7.1)	0.803 ^g^
Tacrolimus ^d^	494 (90.3)	163 (89.1)	166 (91.2)	165 (90.7)	0.773 ^g^

eGFR = estimated-Glomerular Filtration Rate, eGFR–CKD-EPI (mL/min per 1.73 m2); PTH = parathyroid hormone; CNI = calcineurin inhibitors; ^a^ one-year post-transplant data available for 462 patients; ^b^ mean ± standard deviation (SD); ^c^ median (interquartile range); ^d^ number (%); ^e^
*p*-value by student-t test; ^f^
*p*-value by Mann-Whitney *U* test; ^g^
*p*-value by chi-square test; *p*-values in bold indicate statistical significance.

**Table 3 jcm-09-02893-t003:** Number (%) of cardiovascular events and deaths.

	Total (*n* = 547)	Low (*n* = 183)	Medium (*n* = 182)	High (*n* = 182)	*p*-Value
**Median follow-up, years**	3.1 (1.4, 5.2)	3.6 (2.0, 6.0)	2.6 (1.3, 4.9)	2.7 (1.2, 4.2)	-
**Mortality**					
All-cause	80 (14.7)	14 (7.7)	22 (12.1)	44 (24.3)	**<0.001**
Cardiovascular	32 (5.9)	4 (2.2)	10 (5.5)	18 (9.9)	**0.007**
**Cardiovascular events**					
MI	46 (8.4)	7 (3.8)	17 (9.3)	22 (12.1)	**0.015**
CVA	16 (2.9)	6 (3.3)	7 (3.8)	3 (1.6)	0.434
TIA	2 (0.4)	0 (0.0)	0 (0.0)	2 (1.1)	0.134
PAD	30 (5.7)	7 (3.8)	9 (4.9)	15 (8.2)	0.166
Combined	84 (15.5)	18 (9.8)	30 (16.5)	36 (19.8)	**0.027**

Median follow-up after transplantation (interquartile range); cardiovascular events and deaths presented as numbers (%), *p*-value by chi-square test MI = myocardial infarction; CVA = cerebrovascular accident; TIA = transient ischemic attack; PAD = new onset peripheral artery disease, Fontaine III or IV; *p*-values in bold indicate statistical significance.

**Table 4 jcm-09-02893-t004:** Multivariable adjusted associations of CaScore with all-cause mortality, cardiovascular events, and cardiovascular mortality.

	Low	Medium		High			Continuous		
	Hazard Ratio	Hazard Ratio	95% CI	Hazard Ratio	95% CI	P (Trend) Value	Hazard Ratio	95% CI	*p*-Value
**All-cause mortality**									
Univariate	1.0 (Ref)	2.12	1.08–4.15	5.22	2.84–9.58	**<0.0001**	2.02	1.59–2.56	**<0.0001**
Model 1	1.0 (Ref)	2.20	1.12–4.32	5.45	2.94–10.07	**<0.0001**	2.02	1.60–2.56	**<0.0001**
Model 2	1.0 (Ref)	1.61	0.80–3.25	3.40	1.69–6.85	**<0.001**	1.72	1.31–2.28	**<0.001**
Model 3	1.0 (Ref)	1.39	0.67–2.88	3.31	1.60–6.87	**<0.001**	1.75	1.31–2.33	**<0.001**
Model 4	1.0 (Ref)	1.28	0.61–2.70	2.84	1.34–6.00	**0.002**	1.59	1.19–2.14	**0.002**
Model 5	1.0 (Ref)	1.27	0.60–2.66	2.59	1.22–5.51	**0.006**	1.53	1.14–2.06	**0.005**
Model 6	1.0 (Ref)	1.50	0.64–3.51	3.12	1.34–7.26	**0.006**	1.72	1.22–2.43	**0.002**
**Cardiovascular mortality**									
Univariate	1.0 (Ref)	3.42	1.07–10.91	7.86	2.63–23.47	**<0.0001**	2.27	1.55–3.34	**<0.0001**
Model 1	1.0 (Ref)	3.45	1.08–11.05	7.69	2.55–23.14	**<0.0001**	2.24	1.53–3.29	**<0.0001**
Model 2	1.0 (Ref)	2.40	0.72–8.00	4.66	1.38–15.74	**0.009**	1.94	1.24–3.04	**0.004**
Model 3	1.0 (Ref)	1.91	0.56–6.59	4.32	1.22–15.38	**0.014**	1.92	1.19–3.09	**0.007**
Model 4	1.0 (Ref)	1.65	0.47–5.74	4.40	1.19–16.24	**0.013**	2.03	1.20–3.42	**0.008**
Model 5	1.0 (Ref)	1.64	0.46–5.76	4.46	1.19–16.71	**0.014**	2.04	1.20–3.45	**0.008**
Model 6	1.0 (Ref)	2.30	0.64–8.32	55.67	1.68–8.32	**0.004**	2.13	1.32–3.43	**0.002**
**Cardiovascular events**									
Univariate	1.0 (Ref)	2.13	1.19–3.84	2.86	1.62–5.08	**<0.001**	1.56	1.25–1.95	**<0.0001**
Model 1	1.0 (Ref)	2.20	1.22–3.98	2.61	1.46–4.64	**<0.001**	1.60	1.28–1.99	**<0.0001**
Model 2	1.0 (Ref)	1.88	1.00–3.53	2.45	1.25–4.81	**0.010**	1.48	1.13–1.93	**0.004**
Model 3	1.0 (Ref)	1.69	0.89–3.20	2.58	1.20–4.77	**0.013**	1.48	1.12–1.94	**0.005**
Model 4	1.0 (Ref)	1.48	0.76–2.88	2.13	1.05–4.35	**0.034**	1.39	1.04–1.85	**0.025**
Model 5	1.0 (Ref)	1.47	0.76–2.85	1.98	0.97–4.07	**0.060**	1.35	1.01–1.80	**0.042**
Model 6	1.0 (Ref)	1.47	0.76–2.87	2.17	1.08–4.37	**0.028**	1.45	1.09–1.92	**0.010**

Data are presented as hazard ratio and 95% confidence interval (CI) for the tertiles (low, medium, high) and the continuous (natural log transformed) CaScore outcome; *p*-values in bold indicate statistical significance. Model 1: adjusted for transplant center and time between computed tomography and transplantation; Model 2: adjusted for model 1 plus age at time of transplantation, and gender; Model 3: adjusted for model 2 plus BMI, total cholesterol level, diabetes mellitus, smoking status, and systolic blood pressure; Model 4: adjusted for model 3 plus type of dialysis (pre-emptive, hemodialysis, peritoneal dialysis), dialysis vintage pre-transplantation, and number of previous transplantations; Model 5: adjusted for model 4 plus Charlson Comorbidity Index; Model 6: adjusted for model 2 plus one year-post transplant variables (eGFR, calcium, phosphate, albumin, PTH).

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
