# Peer review of "Aorto-Iliac Artery Calcification Prior to Kidney Transplantation"

_jcm, 2020, doi:10.3390/jcm9092893_

Round 1

Reviewer 1 Report

The authors investigated whether a novel non-contrast enhanced CT-based quantification technique for aorto-iliac calcification can be used for cardiovascular risk stratification prior to kidney transplantation and demonstrated an independent association of the aorto-iliac CaScore with all-cause mortality, cardiovascular mortality, and cardiovascular events. While this study is interesting, there are several comments for the authors consideration.

List of Comments to the Author

Major points

#1. Authors should add following covariates in multivariate analysis: calcineurin inhibitor, eGFR (best eGFR after Tx), Serum albumin, C-reactive protein, corrected calcium, phosphate, iPTH, etc. (especially, eGFR)

#2. Because number of outcomes was not large enough (total all-cause mortality; n=80, cardiovascular mortality; n=32, cardiovascular events; n=84), author could not adjust back ground data (over-fitting of the multivariable model). So, authors have to perform another data reduction technique for example propensity score (PS).

Minor points

#1. Agatston score is very skewed data, so you may use log transformed data in the analysis. If the Agatston score is 0, how did you do? (at continuous data analysis).

#2. Figure 3. At interaction analysis, age (p=0.191) and DM (p=0.141) are significant. Because of the underpowered nature of the interaction analysis, we used a 20% significance level for hypothesis testing of the interaction terms*.

*Ramos LF, Shintani A, Ikizler TA, Himmelfarb J. Oxidative stress and inflammation are associated with adiposity in moderate to severe CKD. J Am Soc Nephrol. 2008 Mar;19(3):593-9.

#3. Authors should mention about another part of aortic calcification (and prognosis in Tx) including coronary artery calcification* in the discussion.

* Nguyen PT, Henrard S, Coche E, Goffin E, Devuyst O, Jadoul M. Coronary artery calcification: a strong predictor of cardiovascular events in renal transplant recipients. Nephrol Dial Transplant. 2010 Nov; 25(11): 3773–8.

* Roe P, Wolfe M, Joffe M, Rosas SE. Inflammation, coronary artery calcification and cardiovascular events in incident renal transplant recipients. Atherosclerosis. 2010 Oct; 212(2): 589–94.

Reviewer 2 Report

Comments to the Authors:

Benjamens et al. report on 547 kidney transplant recipients from two institutions in the Netherlands in whom a pretransplant Ct scan including the aorto-iliac region had been performed.  They apply an established computerized methodology for quantitating vascular calcification to the measurement of the calcium score in the aorto-iliac region on the side subsequently used for implanting the transplanted kidney.  They carried out detailed analyses of the correlation of the Calcium score (CaScore) as a categorical and as a continuous variable for the primary endpoint, all cause mortality, and secondary endpoints of cardiovascular mortality and cardiovascular events post transplantation.  They conclude that the CaScore correlates significantly with the primary and secondary endpoints and that  "these findings can improve the identification of patients at risk for early ( cardiovascular death) and those who could potentially benefit from stringent cardiovascular monitoring to improve their prognosis after transplantation".

The approach to evaluating vascular calcification used by the author is both novel and simple.  Surgeons routinely examine the aorto-iliac area pretransplantation looking for possible intraoperative obstacles and complications.  The authors applied an established technology for quantitating coronary artery calcification to the aorto-iliac region and thus brought a reliable, reproducible technique to what is generally a less  stringent assessment.  Herein however lies a weakness of this study in that not all kidney transplant candidates had undergone a CT scan of the aorto-iliac region prior to transplantation.  Instead CT scans were performed only on transplant candidates  meeting certain age and risk criteria.  Thus the cohort study suffers from selection bias and the data resulting from the analysis of this study cannot be generalized to the overall transplant candidate population.

The data presentation and the analytic strategies are generally clear.  However, an infrequently used statistical tool called "integrated discrimination improvement" ( IDI) is used without referencing or explanation .   This needs to be corrected.  Further, another ambiguity in the data presentation is a lack of clarity as to whether or not cardiovascular deaths were included in cardiovascular events. 

Despite the excellent c-statistics correlating the CaScore with the primary and secondary endpoints the authors go on to combine the CaScore with the Charlson Comorbidity Index and the Framingham Risk Score and although the improvement in the c-statistic was not statistically significant they report a significant IDI.   The exact reason for this exercise is not clear since the CaScore alone was an excellent predictor of outcomes and neither of the two additional scores are routinely used or easy to apply.   Importantly many of the risk factors in the Charlson score are already some of the risk factors used to decide whether to perform a CT scan on an transplant candidate- confounding the utility of such an exercise.

The authors state that the CaSCore should be informative regarding "early cardiovascular deaths" post transplantation .  While the authors do not tell us how long after transplantation the deceased patients suffered a cardiovascular death, examination of the Kaplan-Meier curves in Figures 4A and B shows that cardiovascular events and deaths do not appear until about two years post transplantation.   This is not a time period post transplantation conventionally thought of as "early".    In fact it is curious that it takes two years before patients at such high risk begin to experience the anticipated consequences of their underlying disease - a time when immunosuppression has been reduced and steroids often discontinued.  

Finally, unexpectedly and without clear rationale the authors in the discussion launch into a treatise about frailty and post-transplant outcomes.  While this reviewer acknowledges the role of frailty in post-transplant outcomes, there is nothing about the data presented that sheds any light on the presence or absence of frailty in this cohort, and therefore this embellishment of the discussion should be deleted. 
